# Comparison of Four Density-Based Semi-Empirical Models for the Solubility of Azo Disperse Dyes in Supercritical Carbon Dioxide

**Jun Yan [1],\*, Shuang Du [1], Hui Du [1], Huan Zhang [1], Andong Jiao [1], Hong Li [1], Bing Du [1], Dawei Gao [2] and Kaihua Wang [3],\***

1. National Supercritical Fluid Waterless Dyes Technology R&D Center, Liaoning Provincial Key Laboratory of Ecological Textiles, Dalian Polytechnic University, Dalian 116034, China
2. School of Textile and Clothes Engineering, Yancheng Institute of Technology, Yancheng 224051, China
3. School of English, Liaoning Vocational College of Light Industry, Dalian 116100, China
* Correspondence: yanjun@dlpu.edu.cn (J.Y.); kaihuabei@163.com (K.W.)

**Abstract:** Compared to traditional water dyeing, supercritical $CO_2$ fluid waterless dyeing is more advanced concerning zero pollution, energy saving and emission reduction. The measurement of the solubility of disperse dyes in supercritical $CO_2$ provides convenience and technological basis for the popularization and development of this technology. In the current work, the solubility of 2-[4-(2-Cyanoethylethylamino)phenyl]diazenyl-5-nitrobenzonitrile (C.I. Dispersed Red 73), 2,2′-[[4-[(4-Nitrophenyl)azo]phenyl]imino] bis-ethano (C.I. Dispersed Red 19) and 3,3′-[[4-[(4-nitrophenyl)azo]phenyl]imino] bispropiononitrile (C.I. Dispersed Orange S-RL) in supercritical $CO_2$ was determined by flow-type supercritical fluid equipment at pressures ranging from 14 to 26 MPa and temperatures ranging from 343.15 to 403.15 K, and the solubility ranges were $1.96$–$19.78 \times 10^{-6}$, $1.51$–$2.63 \times 10^{-6}$ and $1.49$–$2.49 \times 10^{-6}$ mol/mol, respectively. The increase in pressure P and temperature T has obvious effect on the increase in dye solubility. Dyes with high polarity have low solubility. Four density-based, semi-empirical models—the Chrastil, Mendez-Santiago–Teja, Kumar–Johnston and Bartle models—were employed to correlate and predict the solubility data obtained. The results showed that the relationship between the calculated and experimental values correlated best with the Kumar–Johnston model.

**Keywords:** solubility; supercritical carbon dioxide; azo disperse dye; semi-empirical model; correlation

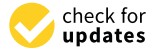



## 1. Introduction

The traditional dyeing process requires much energy consumption and has a high cost; it is difficult to manage environmental pollution, so research into sustainable dye production has received much attention [1]. Since supercritical fluid dyeing technology was developed by researchers in Germany in the late 1980s, many research papers and achievements have been reported from laboratory scale to pilot scale [2–5]. In the textile field, dyeing by supercritical carbon dioxide fluid (sc–$CO_2$) does not need added dispersants, leveling agents, carriers or other dyeing auxiliaries, and at the same time, no dye wastewater or other wastes are generated. The dyed fabrics are in a dry state without drying treatment, which can completely realize clean, green and environmentally friendly processing. In its supercritical condition (critical condition is $P_c = 7.37$ MPa, $T_c = 31.3$ °C), $CO_2$ has diffusivity and permeability similar to gas, and a density similar to liquid at the same time, which provides a necessary prerequisite for the dissolution and dyeing of dyes in sc-$CO_2$. Moreover, $CO_2$ is not only easy to obtain and simple to process, but it also has two symmetric polar bonds of linear nonpolar molecules, which give sc-$CO_2$ nonpolar properties. Therefore, traditional water-soluble dyes cannot obtain satisfactory results in

this gas because of the polar water-soluble groups; thus, only hydrophobic dyes, such as disperse dyes, can be used in this system [6,7]. Disperse dyes with low polarity sc-$CO_2$ have much better solubility than other kinds of dyes [8–14]. For example, the polar hydroxyethyl group lowers the solubility of C.I. Disperse Blue 27 in sc-$CO_2$, but the solubility of C.I. Disperse Red 169 and C.I. Disperse Brown 22 is enhanced by the presence of halogens in sc-$CO_2$ [15]. The azo dyes are the largest and most versatile class of organic dyestuffs, with shades covering the entire spectrum [16]. Moreover, some experiments have proved that the azo disperse dyes have good solubility and superior dyeing effect in sc-$CO_2$ [17–19].

The degree to which the dyestuff is dissolved in sc-$CO_2$ acts as a vital role in the whole dyeing process. In general, the solubility of dyes affects the dye uptake and dyeing quality of fabrics. Therefore, one of the important tasks is to accelerate the industrial application of supercritical waterless dyeing technology to accurately obtain the solubility data for different disperse dyes in sc-$CO_2$ [20–23]. However, concerning industry application, published research and relevant data are lacking. Kong et al. [24] studied the binary and quaternary solubility of C.I. Disperse Orange 30, C.I. Disperse Red 167 and C.I. Disperse Blue 79 and their multicomponent systems in the following sc-$CO_2$ ranges: 0.122–67.9 $\times$ $10^{-7}$, 0.0929–21.2 $\times$ $10^{-7}$, 0.0769–37.2 $\times$ $10^{-7}$ and 0.0739–63.7 $\times$ $10^{-7}$ mol/$m^3$, respectively, with the order of magnitude between $10^{-6}$ and $10^{-9}$. They found that with the increase in pressure, the density of sc-$CO_2$ increases, which leads to an increase in the solubility of the dyestuff. Additionally, higher temperature is conducive to solubility of the dyestuff. Draper et al. [15] investigated the solubilities of ten disperse dyes, including azo disperse dyes and anthraquinone disperse dyes, in sc-$CO_2$ using a modified apparatus. The results showed the difference in solubility of disperse dyes with similar molecular structure in sc-$CO_2$ was related to molecular polarity. Yamini et al. [25] measured the solubilities of C.I. Disperse Yellow 232 and 184 and their modified forms by using the static method, and the solubility data obtained from the experiment were correlated with the Chrastil, Kumar–Johnston (K–J), Bartle and Mendez-Santiago–Teja (MST) semi-empirical models. The results showed that the four models have good consistency in their correlation with experimental data. Additionally, the average absolute relative deviation (AARD) is between 15.5% and 18.4%.

In this work, three azo-based disperse dyes (C.I. Disperse Red 73, C.I. Disperse Red 19 and C.I. Disperse Orange S-RL) were selected to measure the solubility in sc-$CO_2$ by a flow-type experimental apparatus, which are different from the dyes used by Yamini et al. This was performed to explore the solubility trends of different species of disperse dyes in sc-$CO_2$ with changing temperature and pressure, and to correlate and predict solubility data. The parent structure of three disperse dyestuffs is similar, but the positions and types of the substituent group in each dyestuff are different. They can be used for dyeing polyester fibers and other applicable fibers in supercritical $CO_2$ fluid. In order to accommodate the deficient quantity of experimental data, the solubility calculation model was applied to correlate the relationship between experimental conditions and solubility in the measurement process, and the solubility data outside the range of measurement conditions were predicted, so as to obtain more comprehensive and systematic solubility information. After measuring the solubility of the dyes at temperatures from 343.155 to 403.15 K and pressures from 14 to 26 MPa by using the gravimetry method, density-based, semi-empirical models, including the Chrastil [26], MST [27], K–J [28] and Bartle [29] models, were chosen to correlate and estimate the solubility data for the three disperse dyes. These basic solubility data may assist in the proper design of supercritical fluid dyeing processes in industry and expand the scale and impact of supercritical dyeing and its applications.

## 2. Materials and Methods

### 2.1. Materials

C.I. Disperse Red 73 (DR 73), C.I. Disperse Red 19 (DR 19) and C.I. Disperse Orange S-RL (DO S-RL) were purchased from Hebei Province Zize Chemical Co., Ltd., without any additives. The molecular structure, molecular weight (MW) and source of the three dyes

are given in Table 1. $CO_2$ gas was provided by Dalian Zhonghao Guangming Chemical Research and Design Institute Co., Ltd. (Dalian, China), and its purity is above 99.995%. The metal screen mesh used in the experiment is 304 stainless steel, 300 mesh, which is commercially available. All reagents are used as supplies without any treatment.

**Table 1.** CAS number, chemical formula, molecular structure, molecular weight, source and mass fraction purity of the three disperse dyestuffs.

| Dye Name | Formula | Molecular Structure | MW | Melting Point/K | Mass Fraction Purity | Source |
|---|---|---|---|---|---|---|
| C.I. disperse red 73(2-[4-(2-Cyanoethylethylamino) phenyl]diazenyl-5-nitrobenzonitrile) CAS number 16889-10-4 | $C_{18}H_{16}N_6O_2$ | | 348.36 | $402.15 \pm 0.05$ | >0.995 | Hebei Province Zize Chemical Co., Ltd. (Handan, China) |
| C.I. disperse red 19(2,2′-[[4-[(4-Nitrophenyl)azo]phenyl] imino]bis-ethano) CAS number 2734-52-3 | $C_{16}H_{18}N_4O_4$ | | 330.34 | $412.15 \pm 0.06$ | >0.995 | Hebei Province Zize Chemical Co., Ltd.(Handan, China) |
| C.I. disperse orange S-RL (3,3′-[[4-[(4-nitrophenyl)azo]phenyl] imino]bispropiononitrile) CAS number 4234-72-4 | $C_{18}H_{16}N_6O_2$ | | 348.36 | $398.15 \pm 0.03$ | >0.995 | Hebei Province Zize Chemical Co., Ltd.(Handan, China) |
| Carbon dioxide CAS number 124-38-9 | $CO_2$ | — | 44 | | >0.999 | Dalian Zhonghao Guangming Chemical Research and Design Institute Co., Ltd. (Dalian, China) |

### 2.2. Apparatus and Procedures

The measurement process of the entire experiment was determined by a self-regulating dynamic apparatus made in the laboratory. As shown in Figure 1, the high-pressure pump (4) extracts the carbon dioxide gas from a $CO_2$ cylinder (1) that flows to a purifier (2) for purification. After the condenser, the carbon dioxide is converted from the gaseous phase to the liquid phase, and the liquid carbon dioxide fluid is pumped into the dyeing autoclave (9), through the heat exchanger (8) and into the high-temperature and high-pressure dyeing autoclave, where it is fully contacted with the dye; the dye is dissolved in the flow of the supercritical carbon dioxide. The supercritical fluid carrying dye molecules flows from the dyeing autoclave (9) to the separation autoclave (10), and the solute dissolved in sc-$CO_2$ is separated by decompression and cooling and deposited on the bottom of the separation autoclave (10). The pressure in the autoclave is controlled by an automatic back-pressure regulator (11) during operation. When the solution equilibrium time is reached, the operation of the equipment is stopped and the valve to discharge the gas is opened slowly. Mass flow rate and flow of sc-$CO_2$ real-time data from the flowmeter are recorded by the computer software, and the experimental settings, conditions and control of the operating process are monitored by the computer software, which can display real-time parameter data through the main controller. The estimated errors for temperature and pressure measurements are 0.1 K and 0.1 MPa.

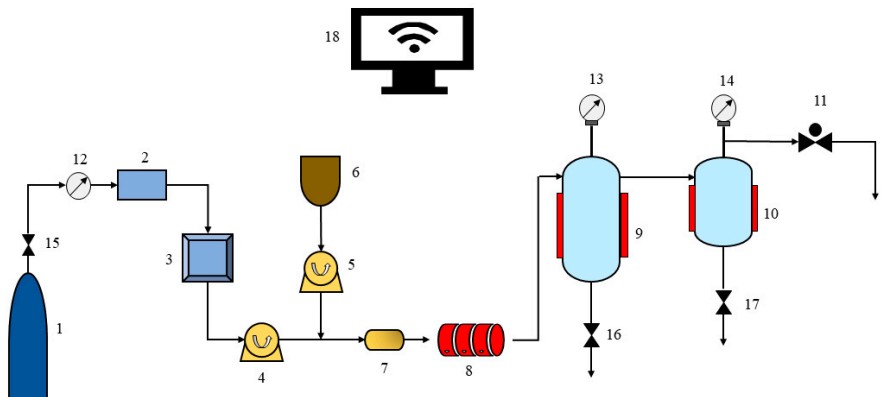

**Figure 1.** Diagram of the apparatus equipped with (1) $CO_2$ cylinder, (2) purifier, (3) refrigerator, (4) high-pressure pump (Model P-50A, Thar SFC, a Waters Co., USA), (5) co-solvent pump, (6) co-solvent tank, (7) mixer, (8) heat exchanger (Model 04424-1, Thar SFC, a Waters Co., USA), (9) dyeing autoclave (500 mL), (10) separation autoclave (200 mL), (11) back-pressure regulator, (12) flow gauge, (13–14) manometer, (15–17) valves.

### 2.3. Measurement of Solubility

In this work, the solubility data for the three dyes were measured at experimental pressures of 14, 18, 22, 24 and 26 MPa and temperatures of 343.15, 363.15, 383.15, 393.15 and 403.15 K, respectively. Preliminary experiments were carried out before the determination. According to the results of the preliminary experiment, the solubility of the dye remained stable after 60 min, so the equilibrium time of the dye dissolution in the experiment was set at 60 min. The solubility measurement steps as follow: Weigh 1.0 g dye and wrap it in a metal screen to prevent leakage. Place the dye package gently at the bottom of the autoclave. The machine was launched to attain the experimental parameters. Reach the preset conditions, and the mass flow rate of $CO_2$ was set at 10 g/min. After 60 min, the dye package was taken out and again accurately weighed on an electronic balance and recorded. In this study, the gravimetry method was used to calculate the solubility of the dyes. The weight mass difference before and after dissolution was the dissolved mass of dyes in sc-$CO_2$ [30]. Each experiment was run in triplicate and the average value was taken as the final result. The experimental average relative standard deviation for DR 73, DR 19 and DO S-RL was 20.08%, 1.37% and 1.93%, respectively. The solubility of the dye in sc-$CO_2$ was expressed in mole fraction ($y$), and the solubility of three dyestuffs was calculated by applying the following equation [4]:

$$y = \frac{n_{dye}}{n_{dye} + n_{co_2}} \tag{1}$$

where $n_{dye}$ is the molar mass of the dyestuff, mol; $n_{dye} = m_{dye}/M_{dye}$, $m_{dye}$ is the dissolved dye mass, g; $M_{dye}$ is the relative molecular mass of the dye, g/mol; $n_{co_2}$ is the mole number of $CO_2$, mol.

## 3. Results and Discussion

### 3.1. Experimental Solubility Data in sc-$CO_2$

Solubility is defined as the mole fraction ($y$) of solute in sc-$CO_2$ in mol/mol. The solubilities of DR 73, DR 19 and DO S-RL in sc-$CO_2$ for different conditions in the experiment are listed in Table 2. The density of sc-$CO_2$ was acquired from the online website of NIST [31]. The solubility of DR 73 was in the range $1.96\text{–}12.38\cdot10^{-6}$ mol/mol, that of DR 19 was in the range $1.51\text{–}2.63\cdot10^{-6}$ mol/mol, and that of DO S-RL was in the range $1.49\text{–}2.45\cdot10^{-6}$ mol/mol. The solubility of the three dispersed dyes increased gradually with the increase in pressure, and the increase in solubility was small when the pressure increased from 14 to 18 MPa. In contrast, the solubility increased sharply from 18 to 26 MPa.

The reason may be that the density of sc-$CO_2$ increases with the increase in system pressure; the distance between the dye molecule and the carbon dioxide molecule decreases and the intermolecular force is strengthened, leading to enhanced solubility of dyestuffs in sc-$CO_2$ [12]. In consequence, the solubility of the dyestuffs at high pressure is larger than that at low pressure. Additionally, the three dyestuffs all dissolved increasingly more as the temperature increased from 343.15 to 403.15 K.

**Table 2.** Solubility data for crystalline DR 73, crystalline DR 19 and crystalline DO S-RL in sc-$CO_2$, and the density of sc-$CO_2$.

| T/K | P/MPa | $\rho$/kg·m$^{-3}$ | $10^6 \cdot y$/mol·mol$^{-1}$ | | |
|---|---|---|---|---|---|
| | | | **DR 73** | **DR 19** | **DO S-RL** |
| 343.15 | 14 | 456.62 [3] | 1.96 [1] ± 0.28 [2] | 1.51 ± 0.15 | 1.49 ± 0.44 |
| | 18 | 612.24 | 2.78 ± 0.17 | 1.58 ± 0.14 | 1.62 ± 0.47 |
| | 22 | 695.1 | 3.98 ± 0.39 | 1.84 ± 0.17 | 1.72 ± 0.11 |
| | 24 | 724.23 | 5.03 ± 0.31 | 1.93 ± 0.24 | 1.79 ± 0.52 |
| | 26 | 748.61 | 7.16 ± 0.39 | 2.08 ± 0.29 | 1.93 ± 0.42 |
| 363.15 | 14 | 335.08 | 4.02 ± 0.24 | 1.55 ± 0.20 | 1.60 ± 0.37 |
| | 18 | 475.67 | 6.05 ± 0.20 | 1.69 ± 0.16 | 1.70 ± 0.67 |
| | 22 | 580.38 | 7.79 ± 0.15 | 1.98 ± 0.13 | 1.76 ± 0.56 |
| | 24 | 619.21 | 9.23 ± 0.18 | 2.03 ± 0.14 | 1.90 ± 0.40 |
| | 26 | 651.66 | 12.39 ± 0.49 | 2.25 ± 0.21 | 2.10 ± 0.50 |
| 383.15 | 14 | 276.12 | 6.06 ± 0.42 | 1.70 ± 0.10 | 1.87 ± 0.53 |
| | 18 | 384.99 | 8.16 ± 0.34 | 1.73 ± 0.22 | 1.94 ± 0.47 |
| | 22 | 484.04 | 10.31 ± 0.29 | 2.10 ± 0.19 | 1.99 ± 0.08 |
| | 24 | 525.8 | 12.38 ± 0.49 | 2.18 ± 0.21 | 2.06 ± 0.33 |
| | 26 | 562.32 | 14.48 ± 0.20 | 2.39 ± 0.37 | 2.16 ± 0.56 |
| 393.15 | 14 | 256.41 | 9.31 ± 0.39 | 1.82 ± 0.16 | 2.01 ± 0.35 |
| | 18 | 353.55 | 11.36 ± 0.40 | 1.84 ± 0.30 | 2.10 ± 0.63 |
| | 22 | 445.86 | 13.87 ± 0.45 | 2.14 ± 0.21 | 2.15 ± 0.38 |
| | 24 | 486.7 | 16.57 ± 0.60 | 2.29 ± 0.20 | 2.23 ± 0.71 |
| | 26 | 523.39 | 17.81 ± 0.28 | 2.54 ± 0.13 | 2.34 ± 0.56 |
| 403.15 | 14 | 240.4 | 12.24 ± 0.20 | 1.88 ± 0.13 | 2.03 ± 0.40 |
| | 18 | 328.23 | 14.55 ± 0.43 | 1.93 ± 0.09 | 2.11 ± 0.31 |
| | 22 | 413.72 | 15.48 ± 0.54 | 2.23 ± 0.17 | 2.23 ± 0.48 |
| | 24 | 452.85 | 17.59 ± 0.22 | 2.38 ± 0.24 | 2.32 ± 0.86 |
| | 26 | 488.81 | 19.78 ± 0.49 | 2.63 ± 0.19 | 2.45 ± 0.74 |

[1] Average solubility for three repeated experiments. [2] Standard deviation of the average solubility. [3] $CO_2$ density is obtained by the Span−Wagner equation of state [32].

In addition to pressure and temperature, which are two important factors, molecular polarity also affects the solubility of dyes. Supercritical $CO_2$ fluid is a nonpolar solvent. According to the principle of similar phase solution, solutes with weak polarity or non-polarity can be dissolved. The stronger the polarity of the dye molecules as solutes, the worse the corresponding solubility [33]. The molecular weight of DR 73 and DO S-RL is the same, and there are two cyanide groups in the molecular structure of DR 73, which is one cyanide group on the benzene ring and the other on the carbon chain. The two cyanide groups of DO S-RL molecule are on the carbon chain. The difference in the position of the cyanide group results in the difference in molecular polarity between the two dyes. From the solubility range of the two dyes, the solubility of DR 73 is much greater than that of DO S-RL, which may indicate that the polarity of the DR 73 molecule is lower than that of the DO S-RL molecule.

### 3.2. Experimental Solubility Data Correlation

Semi-empirical models have been used widely to correlate and extrapolate solubility data for disperse dyes in sc-$CO_2$. The Chrastil model [34] assumes that solute and solvent

molecules interact to form a solvent complex, and an equilibrium state is formed between the complex and solvent. A theoretical model based on supercritical $CO_2$ fluid density is proposed; the model is shown in Equation (2):

$$\ln y = a_1 \ln \rho + \frac{b_1}{T} + c_1 \tag{2}$$

where $y$ is the mole fraction of dyestuff in supercritical fluid, mol/mol; $\rho$ is the supercritical $CO_2$ density, kg/m$^3$; $T$ is the system temperature, K; $a_1$, $b_1$ and $c_1$ are model constants obtained by fitting experimental data.

Inspired by dilute solution theory, the MST model [35] was proposed by Mendez-Santiago and Teja based on the algorithm of Henry constant of solute in supercritical fluid. The relationship between solute solubility and system parameters is shown in Equation (3), where $y$ is the mole fraction of dyestuff in supercritical fluid, mol/mol; $\rho$ is the supercritical $CO_2$ density, kg/m$^3$; $P$ is the pressure of the system, MPa; $T$ is the system temperature, K; $A_1$, $A_2$ and $A_3$ are constants.

$$T \ln(yP) = A_1 + A_2\rho + A_3T \tag{3}$$

Similar to the Chrastil model, the K–J model describes the linear behavior of the semi-logarithmic relationship between solute solubility and sc-$CO_2$ density. As shown in Equation (4), $y$ is the mole fraction of dyestuff in supercritical fluid, mol/mol; $\rho$ is supercritical $CO_2$ density, kg/m$^3$; $T$ is the system temperature, K; $A$, $B$ and $C$ are constants.

$$\ln y = A + \frac{B}{T} + C\rho \tag{4}$$

The Bartle model [36] was proposed by Bartle et al. By introducing reference pressure $P_{ref}$ and reference density $\rho_{ref}$, the relationship between solubility and system temperature, pressure and sc-$CO_2$ density was obtained, as exhibited in Equation (5), where $y$ is the mole fraction of dyestuff in supercritical fluid; $a_0$, $a_1$ and $a_2$ are empirical constants, which can be determined by regression from experimental data; $\rho$ is the density of supercritical carbon dioxide fluid; $P$ and $T$ are the pressure and temperature of the system; $P_{ref}$ is the reference pressure, taken as 0.1 MPa; $\rho_{ref}$ is the reference density, taken as 700 kg/m$^3$.

$$\ln\left(\frac{yP}{P_{ref}}\right) = a_0 + \frac{a_1}{T} + a_2\left(\rho - \rho_{ref}\right) \tag{5}$$

After fitting the solubility data for the disperse dyes with the four semi-empirical models, the values of AARD were used to compare the accuracy of the models. The expression is shown in Equation (6), where $y^{exp}$ is the experimental value, $y^{cal}$ is the calculated value and $N$ is the number of experimental data. The smaller *AARD* is, the better the accuracy of model association.

$$AARD\% = \frac{100}{N} \sum_{n=1}^{N} \left| \frac{y_n^{exp} - y_n^{cal}}{y_n^{exp}} \right| \tag{6}$$

Figure 2 shows that the linear regression relationships between the logarithms of solubility for the three disperse dyes and the logarithms of supercritical $CO_2$ density, which were obtained by fitting the measured data with the Chrastil model. As shown in Figure 2a–c, the calculated solubility values of the three dispersed dyes are in good agreement with measured values at temperatures from 343.15 to 403.15 K.

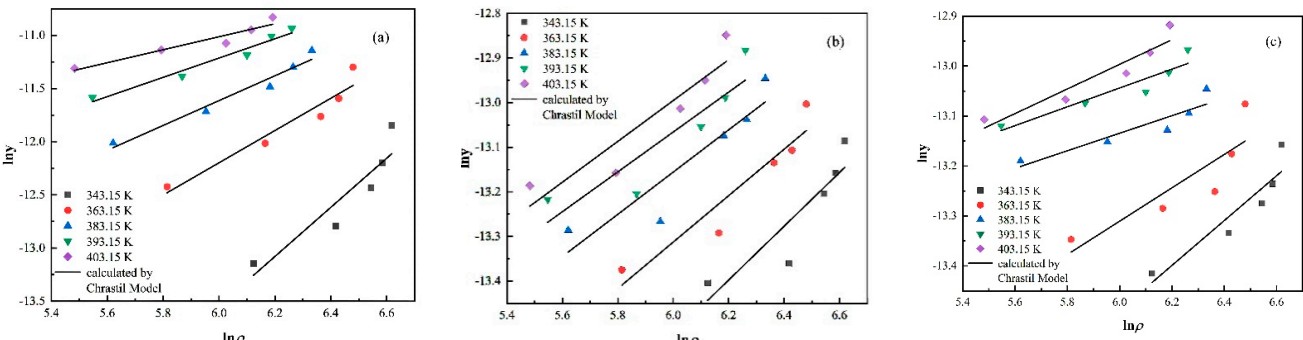

**Figure 2.** Solubilities in sc−CO$_2$ calculated by the Chrastil model for (**a**) DR 73, (**b**) DR 19, (**c**) DR DO S-RL.

Figure 3 depicts a simple linear relationship between Tln (yP)−A$_3$T and supercritical CO$_2$ density ($\rho$) obtained by fitting the MST model expression. It can be seen that the experimental points of DR 73, DR 19 and DO S−RL are uniformly distributed near the fitting line, and the MST model can be used to perform a linear fitting process for the experimental range under the conditions of temperature from 343.15 to 403.15 K and pressure from 14 to 26 MPa.

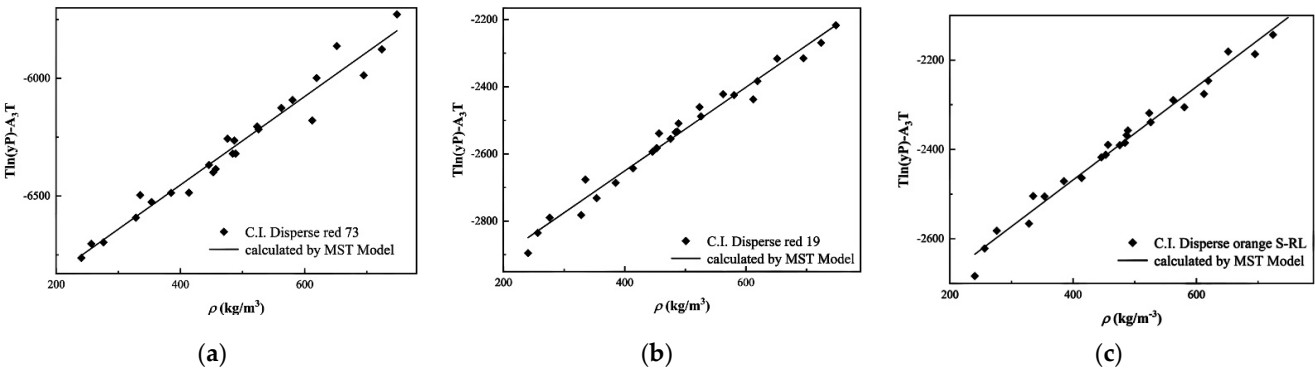

**Figure 3.** Solubilities in sc−CO$_2$ calculated by the MST model for (**a**) DR 73, (**b**) DR 19, (**c**) DO S−RL.

As shown in Figure 4, solubility correlation lines of the three disperse dyes presented after K−J model correlation are similar to the fitting effect of the Chrastil model, which describes the positive correlation between the solubility values of dyestuffs and the density change in sc−CO$_2$.

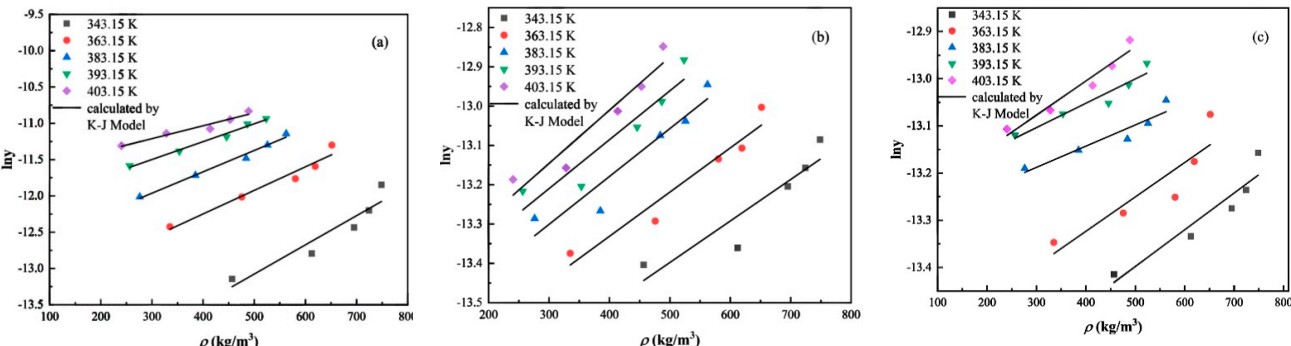

**Figure 4.** Solubilities in sc−CO$_2$ calculated by the K−J model for (**a**) DR 73, (**b**) DR 19, (**c**) DO S−RL.

Figure 5 depicts the fitting linear relationship of the experimental solubility data for DR 73, DR 19 and DO S–RL with the Bartle model. As shown in Figure 6a–c, the calculated values of the solubility agree well with the experimental values.

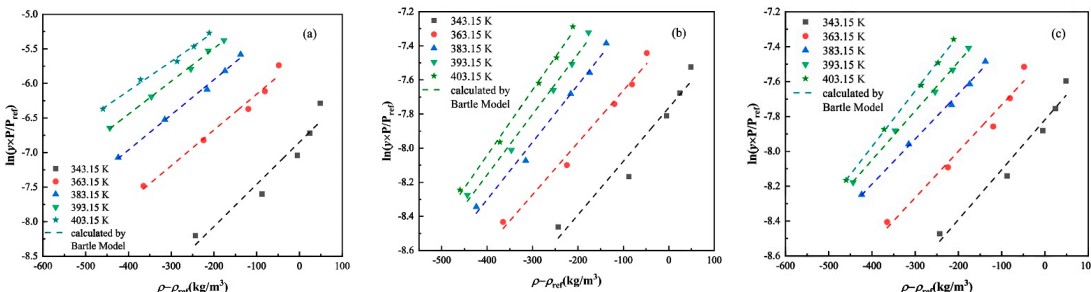

**Figure 5.** Solubilities in sc−CO$_2$ calculated by the Bartle model for (**a**) DR 73, (**b**) DR 19, (**c**) DO S−RL.

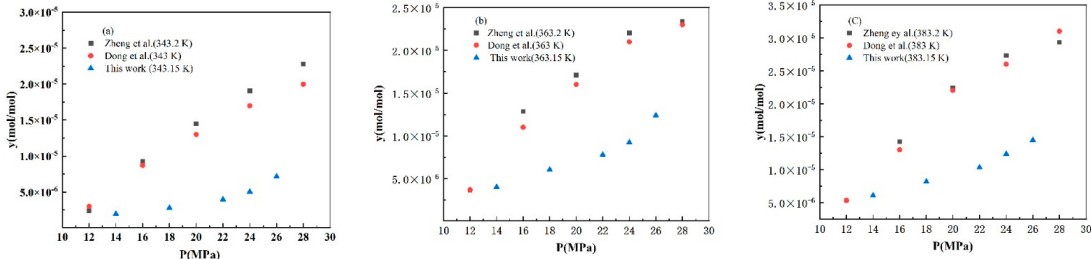

**Figure 6.** Comparison of solubility of DR 73 at (**a**) 343.15 K, (**b**) 363.15 K, (**c**) 383.15 K.

The parameter values of DR 73, DR 19 and DO S-RL acquired by correlation calculation of the four semi-empirical models are listed in Table 3. As summarized in Table 3, the values of AARD% obtained by correlating the solubility of DR 73 in sc−CO$_2$ with the Chrastil model, MST model, K–J model and Bartle model are 12.29, 10.72, 10.37 and 10.51%, respectively. The AARD% of solubility of DR 19 in sc−CO$_2$ correlated with the four models is in the range 3.49 to 5.49%, and DO S-RL is in the range 2.49–3.53%. Among the fitting results of the three disperse dyes, the K–J model has the lowest AARD% value, and the calculated value is closest to the experimental value. Therefore, the K–J model has higher computational accuracy and better correlation effect than the other models. In addition, DR 19 and DO S−RL have much higher accuracy than DR 73 through correlation and extrapolation of the semi-empirical models.

**Table 3.** Model parameters for DR 73, DR 19 and DO S–RL in Equations (2)–(5) and calculated values of AARD%.

| Model | Parameters | Dyestuff Name | | |
|---|---|---|---|---|
| | | **DR 73** | **DR 19** | **DO S-RL** |
| Chrastil | $a_1$ | 1.18 | 0.49 | 0.26 |
| | $b_1$ | −4661.72 | −1087.52 | −967.42 |
| | $c_1$ | −6.45 | −13.23 | −12.15 |
| | AARD (%) | 12.29 | 4.34 | 3.24 |
| MST | $A_1$ | −7203.86 | −3148.06 | −2885.14 |
| | $A_2$ | 1.88 | 1.24 | 1.04 |
| | $A_3$ | 8.10 | −3.37 | −3.81 |
| | AARD (%) | 10.72 | 5.49 | 4.17 |

**Table 3.** *Cont.*

| Model | Parameters | Dyestuff Name | | |
| --- | --- | --- | --- | --- |
| | | **DR 73** | **DR 19** | **DO S-RL** |
| K-J | A | 0.24 | −10.50 | −10.68 |
| | B | −4994.93 | −1203.16 | −1043.50 |
| | C | 0.00295 | 0.00119 | 0.000651 |
| | AARD (%) | 10.37 | 3.49 | 2.49 |
| Bartle | $a_0$ | 11.55 | −0.43 | −0.98 |
| | $a_1$ | −6299.55 | −2507.77 | −2348.12 |
| | $a_2$ | 0.0051 | 0.00333 | 0.0028 |
| | AARD (%) | 10.51 | 4.42 | 3.53 |

*3.3. Comparison Solubility with Literature Data*

To verify the accuracy of solubility measurements in this work, experimental data were compared with those previously reported in the literature. Literature review indicates that only the solubility data for DR 73 in sc-$CO_2$ have been reported. As shown in Figure 6, the variation tendency of solubility experimental data obtained in this work is not consistent with the research results of Zheng et al. [37] and Dong et al. [38]. In addition, the experimental equipment and determination methods are the same in both published reports. AARD values obtained by model fitting reported in the literature and those obtained by the same model fitting in this work are shown in Table 4.

**Table 4.** AARD values fitted by the same model in this work and previous literature.

| Model | This Work | Zheng et al. [37] | Dong et al. [38] |
| --- | --- | --- | --- |
| Chrastil | 12.29 | 5.29 | 9 |
| MST | 10.72 | 17.49 | 13 |

Based on the data from the two published reports, the solubility data for DR 73 obtained under the same experimental apparatus, measurement method and experimental operation correlated with two different empirical models, but the correlated effects showed different changes.

In this work, the correlation effect of the solubility data obtained by different dye quality, equilibrium time, mass flow rate and solubility measurement methods are used; the AARD value for this work was larger than the reported value, based on fitting by the Chrastil model, while the AARD value of the MST model was smaller than the one reported in the literature. This also shows that the solubility of the dye is not affected by the experimental apparatus and measurement method. The solubility data obtained by different measurement apparatus and measuring methods reported in the literature are still useful for reference. From the data comparison information shown in Figure 6, it can be seen that the solubility of DR 73 has a consistent change trend with pressure and temperature. Moreover, the AARD value differs from that reported in the literature because the supercritical carbon dioxide equipment is nonstandard, and different temperatures, pressures and equipment affect the experimental results. Thus, the differences with those reported in the literature do not affect the relationship between solubility and process measurement condition, and is therefore suitable for the exploration of the universal law applicable to disperse dyes in sc-$CO_2$ in supercritical fluid dyeing processes. Therefore, our experimental method is feasible.

**4. Conclusions**

The solubility of C.I. Dispersed Red 73, C.I. Dispersed Red 19 and C.I. Dispersed Orange S-RL in supercritical $CO_2$ fluid was measured using flow apparatus at temperatures from 343.15 to 403.15 K and pressures from 14 to 26 MPa. The solubility data were correlated

and predicted with semi-empirical models, including the Chrastil, MST, K–J and Bartle models. It was found that the solubility increased with the increase in pressure, and the higher temperature was conducive to the dissolution of the dyestuff. The correlation effects of the four semi-empirical models based on the density of supercritical $CO_2$ were compared. Compared to the other three models, the K–J model had a lower value of AARD, ranging from 2.49 to 10.37%, which was more suitable for fitting the solubility of these three azo disperse dyes. Moreover, the solubility of C.I. Dispersed Red 19 and C.I. Dispersed Orange S-RL in supercritical $CO_2$ is more suitable for correlation and extrapolation by semi-empirical models. The measurement of solubility may promote the research progress and industrial application of supercritical carbon dioxide waterless dyeing.

**Author Contributions:** Conceptualization, J.Y. and K.W.; methodology, J.Y.; validation, J.Y., S.D. and H.D.; formal analysis, H.Z.; investigation, H.L. and B.D.; data curation, J.Y.; writing—original draft preparation, J.Y.; writing—review and editing, S.D., H.D. and A.J.; supervision, D.G.; project administration, J.Y. and K.W. All authors have read and agreed to the published version of the manuscript.

**Funding:** This research was funded by pen Project Program of Key Laboratory of Eco-textiles, Ministry of Education, Jiangnan University, grant number KLET1808; Basic Scientific Research Project of the Education Department of Liaoning Province (Key Project); Dalian High-Level Talents Program, grant number 2018RQ26; Foreign Expert Project, grant number No. G20190208012.

**Data Availability Statement:** The data generated in this study is available upon request.

**Acknowledgments:** This work was supported by the Open Project Program of Key Laboratory of Eco-textiles, Ministry of Education, Jiangnan University (No. KLET1808); Basic Scientific Research Project of the Education Department of Liaoning Province (Key Project); Dalian High-Level Talents Program (No. 2018RQ26) and Foreign Expert Project (No. G20190208012).

**Conflicts of Interest:** The authors declare no conflict of interest.

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
