# Peer review of "Comparison of Four Density-Based Semi-Empirical Models for the Solubility of Azo Disperse Dyes in Supercritical Carbon Dioxide"

_processes, doi:10.3390/pr10101960_

Round 1
Reviewer 2 Report
Comparison of four density-based semi-empirical models for the solubility of azo disperse dyes in supercritical carbon dioxide
Supercritical carbon dioxide looks like an interesting research pathway to follow, and the authors have contributed a reasonable effort to gather the scientific data. However, there are a few major suggestions that I would like to emphasize.
1) Authors should consider describing more about the novelty of the current study over the previous publications. (include a description in the introduction)
2) I suggest the authors include the physical and chemical characterization of the supercritical carbon dioxide and the importance of this by emphasizing a few practical applications.
3) Describe in the introduction the importance of solubilization of azo disperse dyes in supercritical carbon dioxide over the water solubilized dyes.
4) Figures need to be more sharpened to make them more visible.
5) Because this is a research paper, I would suggest that the authors should not compare their data with the previous publications. This damages the novelty of the current study. Try to be very brief in this comparison and emphasize your data and its importance more than the previously published data.
Reviewer 3 Report
The work is recommended for publication . As the manuscript has noval formulation and new ways for researchers
Author Response
Dear reviewer,
We greatly appreciate your endorsement of our manuscript and wish you all the best.
Sincerely
Yan Jun
September 18, 2022
Round 2
Reviewer 2 Report
The authors have taken reasonable efforts to work on the reviewers' comments to enhance the quality of the manuscript. Thus, I recommend accepting the manuscript at this stage. Wish you all the very best for your future.